# Role of Histone Tails and Single Strand DNA Breaks in Nucleosomal Arrest of RNA Polymerase

**DOI:** 10.3390/ijms24032295

**Published:** 2023-01-24

**Authors:** Nadezhda S. Gerasimova, Nikolay A. Pestov, Vasily M. Studitsky

**Affiliations:** 1Institute of Gene Biology, Russian Academy of Sciences, 119334 Moscow, Russia; 2Faculty of Biology, Lomonosov Moscow State University, 119234 Moscow, Russia; 3Department of Pharmacology, Rutgers Robert Wood Johnson Medical School, Piscataway, NJ 08854, USA; 4Fox Chase Cancer Center, Philadelphia, PA 19111, USA

**Keywords:** DNA damage, DNA loop, chromatin structure, nucleosome, single-strand DNA breaks, transcription-coupled DNA repair

## Abstract

Transcription through nucleosomes by RNA polymerases (RNAP) is accompanied by formation of small intranucleosomal DNA loops (i-loops). The i-loops form more efficiently in the presence of single-strand breaks or gaps in a non-template DNA strand (NT-SSBs) and induce arrest of transcribing RNAP, thus allowing detection of NT-SSBs by the enzyme. Here we examined the role of histone tails and extranucleosomal NT-SSBs in i-loop formation and arrest of RNAP during transcription of promoter-proximal region of nucleosomal DNA. NT-SSBs present in linker DNA induce arrest of RNAP +1 to +15 bp in the nucleosome, suggesting formation of the i-loops; the arrest is more efficient in the presence of the histone tails. Consistently, DNA footprinting reveals formation of an i-loop after stalling RNAP at the position +2 and backtracking to position +1. The data suggest that histone tails and NT-SSBs present in linker DNA strongly facilitate formation of the i-loops during transcription through the promoter-proximal region of nucleosomal DNA.

## 1. Introduction

Single-strand breaks (SSBs, nicks) are common DNA lesions generated during various processes of cell metabolism (DNA repair by base or nucleotide excision pathways, topoisomerase action, spontaneous hydrolysis, exposure to reactive oxygen species and other free radicals, etc.) [1,2,3,4]. They represent nearly three quarters of all DNA damages arising daily in a mammalian cell [1]. Unrepaired SSBs can block different DNA transactions (transcription, replication and repair) and induce accumulation of double-stranded DNA breaks (DSB). This results in the disruption of cellular metabolism and increased genomic instability and can lead to apoptosis and cancer [4]. Genetic dysfunction of proteins involved in SSB repair (SSBR) cause hereditary diseases (reviewed in [1,4,5,6,7]), including neurological disorders [4,5,8,9].

In most cases SSBs are modified and therefore are not available for immediate ligation [3,4]. Usually nicks are recognized by poly [ADP-ribose] polymerase enzymes PARP1 and PARP2, or by AP endonuclease 1 (APE1) [10], and then repaired through SSBR pathway [11,12,13,14,15,16] or via homology-directed repair, HDR [17].

In eukaryotic cells, nuclear DNA is organized into a chromatin—a complex of nucleic acids and proteins with a nucleosome as a basic unit. The nucleosome consists of a 147-bp DNA fragment wrapped around the octamer of core histone proteins [18] and limits access of DNA repair proteins to DNA. SSBs in nucleosome-free regions are recognized and removed much faster than from nucleosome-covered DNA [19]. Some otherwise undetectable SSBs present in nucleosomal DNA can be recognized by processive enzymes progressing along DNA. Thus, SSBs localized on template DNA strand inhibits progression of eukaryotic RNA polymerase II (Pol II), RNA polymerase (RNAP) of *E. coli* and RNAPs of bacteriophages SP6 and T7 in vitro and in vivo [20,21,22]. Stalled Pol II serves as a signal for the transcription-coupled nucleotide excision repair process (TC-NER) enabling the repair of Pol II-interfering SSBs by the transcription-coupled pathway [23]. SSBs localized on non-template DNA strand (NT-SSBs) do not affect the transcription of histone-free DNA in vitro [24], although they are efficiently repaired in vivo [23,25,26].

We have shown that NT-SSBs can also induce arrest of Pol II and *E. coli* RNAP during transcription in vitro when DNA is organized into a nucleosome [27]. In the presence of NT-SSBs in nucleosomal DNA, Pol II is stalled when the active center of the enzyme is localized 7–17 bp downstream of the NT-SSB [27,28]. We have shown that arrest of the enzyme likely occurs due to formation of intranucleosomal loops (i-loops)—intermediates in which the DNA interacts with histone octamer both in front and behind transcribing RNAP and forms an intranucleosomal DNA loop (i-loop) with the enzyme arrested in the loop [27,29]. Structural analysis of elongation complex formed when the active center of RNAP reaches position +24 bp from the nucleosome boundary [27] demonstrated formation of i-loop intermediate with the enzyme backtracked by ~2–4 bp and arrested within the i-loop [30]. A single NT-SSB at the position +12 bp from the nucleosome boundary strongly stabilizes the i-loop [30]. 

According to the proposed model, an NT-SSB positioned behind the transcribing RNA polymerase facilitates i-loop formation and stabilizes the loop, that in turn induces arrest of the enzyme in the loop and thus interferes with further transcript elongation (Figure 1A). Therefore, i-loops could induce arrest of Pol II during transcription of damaged DNA and possibly serve as a signal for Pol II to detect a damage through a chromatin-specific mechanism [27,29,31]. As Pol II transcribes the majority of eukaryotic protein-coding genes and the DNA in the coding regions of genes transcribed at a moderate level remains organized into nucleosomes [32], the observed effect of NT-SSBs on transcription could be a part of novel chromatin-specific mechanism that allows the detection of NT-SSBs by the transcribing enzyme.

NT-SSBs can be used as an instrument to reveal the regions of the nucleosomal DNA where the i-loops can be formed during transcription. Here we examined the early transcribed region of nucleosomal DNA (+1 to +15 bp in the nucleosome) and evaluated the role of extranucleosomal NT-SSBs and histone tails in arrest of RNAP and formation of the i-loops. Progression of RNAP through this region is sensitive to NT-SSBs localized in the linker DNA and histone tails play an important role in nucleosome-specific pausing and arrest of RNAP. Using a DNA footprinting assay, we also demonstrated formation of an i-loop after stalling RNAP at the position +2. Overall, the data suggest that formation of the transient i-loops can occur during transcription through the promoter-proximal region of the nucleosomal DNA, providing an opportunity for regulation of transcription and possibly for more efficient recognition of DNA damages by the transcribing enzyme.

## 2. Results

### 2.1. Experimental Approach

The process of transcription through chromatin containing a single-strand DNA break or nucleotide gap in non-template DNA strand (NT-SSB or NT-SSG, respectively) was investigated using uniquely positioned mononucleosomes formed on the high-affinity 603 DNA sequence [33,34] (Figure 1B). Positioned nucleosomes present a polar barrier to transcription by Pol II [35], therefore 603 nucleosomes were utilized in permissive transcriptional orientation that better recapitulates the essential characteristics of transcription through chromatin in vivo, e.g., survival of histones H3/H4 and displacement of histones H2A/H2B during transcription [31,36]. To study transcription through nucleosomes containing singe, uniquely positioned DNA damage, the templates were prepared by annealing and subsequent ligation of synthetic oligonucleotides (for gaps at the positions −12, −11 and −10 relative to the promoter-proximal boundary of the nucleosome) or by enzymatic nicking (in case of nick at the position +2). The role of histone tails was evaluated using nucleosomes containing trypsin-cleaved histone octamers. The nucleosomes contained less than 5% of contaminating histone-free DNA (Figure 1C).

The experiments were conducted using *E. coli* RNA polymerase (RNAP) as a convenient experimental model. This enzyme is structurally similar with Pol II and uses the Pol II-type mechanism of transcription through chromatin in vitro [31,35,37]. Although this experimental system has certain limitations [36], it recapitulates the nucleosome-specific arrest of Pol II on the DNA containing NT-SSBs [27]. This approach allows avoiding the step of ligation of Pol II elongation complexes to the nucleosome during in vitro transcription [35] and greatly facilitates structural analysis of the elongation complexes [30]. To initiate transcription, a strong bacterial promoter T7A1 was placed before the nucleosome (Figure 1B).

The nucleosomes were transcribed in vitro (Figure 1B) using a previously developed approach [27]. First, radioactive pulse labeling of RNA was conducted during transcription in the presence of a limited set of ribonucleotide triphosphates (rNTPs) (Figure 1B). The enzyme was stalled at the position −39 (position of the active center of the enzyme relative to the promoter-proximal nucleosomal boundary, designated as EC-39) to synchronize the transcribing RNA polymerase complexes (Figure 1B). Next, the transcript was elongated in the presence of all rNTPs at various concentrations of KCl in the transcription buffer. The concentration of KCl in reactions changes the ionic strength of the solution, which in turn affects the strength of DNA-histone interactions and the height of the nucleosomal barrier to transcription. RNA products were analyzed by denaturing PAGE.

### 2.2. NT-SSB at the Position +2 Induces Arrest of RNAP at the +(10–20) Region

Earlier we found that NT-SSBs at the positions +7, +12 and +17 induce strong arrest of RNAP and yeast Pol II (for NT-SSB+12) when the active center of the enzyme reaches position +24 and backtracks to +20 [27,28,30]; the arrest is accompanied by formation of an i-loop [30]. To identify if NT-SSBs localized closer to the nucleosomal boundary can also induce the enzyme arrest, we examined the template containing NT-SSB at the position +2 (NT-SSB+2) (Appendix A). Transcription of histone-free DNA was highly efficient; only minor pausing of RNAP with minimal difference between the intact and damage-containing templates was observed (Appendix A). Transcription of the nucleosomes with and without NT-SSB+2 in vitro revealed that the break interferes with progression of the enzyme through the +(10–20) region (Figure 2A,B)—a well-known nucleosome-specific pausing site for Pol II in vivo and in vitro [35,38,39]; the effect is most apparent at the lower ionic strength (40 and 150 mM KCl). In contrast to the previously described discrete positions of the arrest (+24, +34 and +44 [27]), in this case RNAP progression is inhibited at several positions within the +(10–20) area. 

According to our previous data, NT-SSB+7 induces arrest of RNAP at the position +24 and does not affect the progression of the enzyme through the +(10–20) region [28]. NT-SSB+2, on the contrary, minimally affects the enzyme pausing at the position +24 (Figure 2A,B) while inhibiting transcription of the +(10–20) region. Thus, our data (Figure 2) revealed a new mechanistically distinct region of RNAP pausing dependent on the presence of NT-SSB localized close to the boundary of nucleosomal DNA. As the most likely mechanism of the nucleosome-dependent arrest of RNAP downstream of the NT-SSB is the formation of an i-loop [27,30], the data suggest that i-loops are likely formed during transcription of the +(10–20) region (Figure 2C).

### 2.3. Extranuclesomal NT-SSBs Can Induce Nucleosome-Specific Arrest of RNAP

Since NT-SSB localized very early in the nucleosome can induce nucleosome-specific arrest of RNAP, the effect of extranucleosomal NT-SSBs on nucleosome-specific arrest of RNAP was studied next. DNA breaks localized ~12 bp upstream of the arrest site have the strongest inhibitory effect on transcription [27]; therefore, pausing/arrest at the first nucleotide of nucleosomal DNA (+1) is expected to be strongly affected by a nick or gap at the positions −(12–10). The DNA templates having single nucleotide gaps in the non-template DNA strand (NT-SSG) at different positions within the −(12–10) region were constructed using annealing and ligation protocols (Appendix A). Transcription of histone-free DNA by *E. coli* RNAP revealed a difference between transcription of intact and damaged templates at the region localized near the damage and near the position +15, respectively (Appendix A). The observed difference might be associated with the gap itself or with technical features of the annealing and ligation protocol and needs further investigation. However, the observed arrest occurred only on a small fraction of the templates and NT-SSGs did not affect the progression of RNAP through the beginning of the nucleosomal DNA (region +1); therefore, this template can be used as a model to investigate the nucleosomal arrest of RNAP.

On the nucleosomal template, the gaps induced arrest of the enzyme at several slightly different positions within DNA region extending from −1 to +3 (Figure 3). The data suggest that an i-loop can possibly form when the RNAP reaches the nucleosomal boundary. According to our previous model [27,29], the DNA-binding surface of the histone octamer should be exposed to form the i-loop, but at the position +1 no DNA-binding surface on the octamer is available to interact with the linker DNA. Therefore, it seems possible that the histone tails can participate in formation of the i-loops.

### 2.4. Histone Tails Increase the Efficiency of the Early Nucleosomal Pausing/Arrest

Our earlier findings revealed the role of histone tails in facilitating the early nucleosomal pausing [40,41]. Thus, removal of core histone tails relieves pausing at position +15 and allows further progression by yeast and human Pol II on the nucleosomal template. Since nucleosomal pausing and i-loop formation are likely connected, the possible role of histone tails in nick-induced RNAP arrest during transcription was evaluated next. Nucleosomes without histone tails were prepared using trypsin cleavage of the donor chromatin [42] and transcribed with and without NT-SSB or NT-SSG present in nucleosomal DNA (Figure 4).

As expected, removal of histone tails resulted in a significant reduction of pausing/arrest of RNAP at least up to position +20 during transcription of nucleosomes without DNA damage (Figure 4A). Removal of histone tails resulted in decreased pausing at +(4–5) and in the +(13–19) regions of nucleosomes without NT-SSB; in the presence of NT-SSB +2, only pausing at +(4–5) was strongly decreased, while pausing at +(13–19) region was slightly decreased (Figure 4A). Similar effect of histone tails on +1 pausing was observed in the case of the template having NT-SSG-11.

To further evaluate the effect of histone tails on damage-induced arrest of RNAP, time-courses of transcription through NT-SSG-11 nucleosomes were conducted (Figure 4C). A new site of RNAP arrest (at −3) induced by the histone tails and by the DNA damage was detected, extending the range of the affected pausing/arrest sites from (−2 + 2) to (−3 + 2). The data also revealed transient pausing of RNAP at the position +1 on intact nucleosomes.

Overall, histone tails increase the efficiency of RNAP pausing/arrest in intact nucleosomes to lower extent than on templates without DNA damages. This effect is most likely explained by formation of i-loops at the positions of RNA polymerase +(13–19) even in the absence of histone tails, while formation of i-loops at the positions +(1–5) is highly dependent on the presence of histone tails. Most likely, histone tails stabilize otherwise unstable i-loops that are formed when RNA polymerase is entering a nucleosome. 

Importantly, the removal of histone tails and the presence of a DNA damage do not affect the position of the RNAP pausing/arrest (Figure 4). In other words, the preferred sites of pausing in the absence of DNA damage and sites of arrest in the presence of NT-SSB or NT-SSG are identical. Since at least in one case an i-loop is formed and induces arrest of RNA polymerase in the presence of NT-SSB on nucleosomal DNA [30], the data suggest that nucleosomal pausing and arrest occur by the same mechanism, namely, through i-loop formation, as was proposed earlier [27,30].

### 2.5. Histone Tails Contribute to Protection of Nucleosomal and Linker DNA from Hydroxyl Radicals

The expected ability of histone tails to interact with extranucleosomal DNA regions was tested using DNA footprinting by hydroxyl radicals (Figure 5). Nucleosomes formed on intact DNA with or without histone tails were incubated in the presence of hydroxyl radicals. As expected, in both cases, regardless of the presence of histone tails, a nucleosome-specific pattern of characteristic 10-bp periodic sensitivity to hydroxyl radicals was observed within core nucleosomal DNA. The strongest protection of DNA from the hydroxyl radicals was observed for DNA regions facing the histone octamer surface. In both intact and tailless nucleosomes, a region of additional protection of adjacent 15-bp linker DNA region was observed (Figure 5). However, in the case of tailless nucleosomes, the protection was much less pronounced (Figure 5), suggesting that histone tails strongly contribute to protection of the linker DNA from hydroxyl radicals.

### 2.6. Formation of an i-loop Containing Backtracked RNAP at the Position +1

To evaluate whether an i-loop can form when the RNAP reaches the boundary of nucleosomal DNA, a template allowing formation of the elongation complexes with RNAP stalled at the position +2 (EC+2) during transcription in the presence of a limited set of NTPs was constructed (Appendix A). At the first step, EC-5 was formed, and then a limited set of NTPs was used to allow progression of RNAP to the position +2. The EC-5 was used as a control. The EC+2 was formed on DNA and nucleosomes containing intact DNA, and DNA footprinting was conducted using hydroxyl radicals.

EC-5 and EC+2 have clearly distinct footprints extending from −12 to +8 and from −8 to +15 on the histone-free DNA template (Figure 6A), confirming progression of the RNAP along the template to the expected positions. When these complexes were formed on the nucleosome, a region of additional protection of the extranucleosomal DNA upstream the enzyme position was detected in the EC+2 complex (−(30 −15) region, Figure 6A), as compared with nucleosomes and EC-5. Since this region extends well beyond the DNA sequence protected by RNA polymerase (Figure 6A) and because it is similar in location and the size of the protected area with the region upstream of RNA polymerase in previously analyzed i-loop [30], this protection most likely occurs due to DNA-histone interactions upstream of the RNA polymerase arrested in EC+2. Thus, the data suggest that i-loop is formed in EC+2 but not in EC-5 (Figure 6B). The protection of the −(30 −15) region upstream of RNA polymerase is incomplete (Figure 6A) and could be somewhat stronger in the EC+2 complex formed in nucleosomes containing NT-SSB-12 where the enzyme is more efficiently arrested at the position +2 during ongoing transcription (Figure 3).

As was shown earlier, the RNAP arrested in the i-loop can backtrack along DNA [30]. To evaluate the actual position of the active center of RNAP in the EC+2, the elongation complex was incubated in the presence of the GreB transcription factor (the experimental approach is shown on Appendix A), as was done previously in the case of the EC+48 and EC+24 arrested in the nucleosome [30,43]. GreB stimulates endonuclease activity of *E. coli* RNAP and induces cleavage of RNA at the position of the active center. Incubation in the presence of GreB revealed backtracking of the enzyme in EC+2 by 1–5 nucleotides to positions −4–+1, with the majority of RNAP stalled at the position +1 (Figure 6C).

In summary, EC+2 exhibits at least two properties expected for an EC containing an i-loop [30]—the region of additional protection of extranucleosomal DNA behind the enzyme detected by the footprinting and RNAP backtracking.

## 3. Discussion

In summary, our data suggest that different single NT-SSBs and NT-SSGs localized within the region extending from −12 to +2 bp induce arrest of RNAP during transcription of the region from −3 to +19 bp in the nucleosome (Figure 2 and Figure 3). For each single DNA damage, a heterogeneous set of closely spaced sites of RNAP arrest centered in +1 and +14 was observed, suggesting that somewhat heterogeneous i-loops are formed there (Figure 2 and Figure 3). The presence of N-terminal tails of core histones results in an increase of the efficiency of RNAP pausing/arrest in the presence of DNA breaks to lower extent than on templates without DNA damages (Figure 4), suggesting that formation of the i-loops at these regions is highly dependent on the presence of histone tails (Figure 7). The footprinting data suggested that the histone tails strongly protect the linker DNA from hydroxyl radicals in intact nucleosomes (Figure 5) and therefore can interact with extranucleosomal DNA and are potentially available for i-loop formation. Finally, our footprinting data established i-loop formation when RNAP reaches position +2 bp in the nucleosome; as expected, the stalled complex contains RNAP backtracked to the primary position +1 (Figure 6).

The positions of DNA damage-dependent RNAP arrest sites determined in this and our previous [27] studies are the following: +2, +14, +24, +34 and +44 bp in the nucleosome. It was previously established that the observed 10 bp periodicity of RNAP arrest in the nucleosome reflects the preferred sites of i-loop formation when the DNA kink introduced by transcribing RNAP directs the DNA behind the enzyme towards the histone octamer [27,30]. The +14 arrest site fits this scenario, while the +1 arrest site does not fit this periodic pattern. However, our previous and current data indicate that due to RNAP backtracking the actual i-loops with known structure are formed at the positions +20 [30] and +1 (Figure 6). Therefore, the data in combination suggest that actual approximate periodicity of RNAP arrest after backtracking is +1, +11, +20, +30 and +40 bp in the nucleosome. It was previously shown that the DNA geometry in the EC+20 is highly preferred for i-loop formation [30]. 

The requirement for N-terminal tails of core histones during i-loop formation in the early transcribed region of nucleosomal DNA is most likely explained by the formation of unstable loops during transcription of the region from −3 to +19 bp in the nucleosome. Since DNA is only minimally uncoiled from the histone octamer during transcription of this region and the surface of the octamer is minimally available for interaction with the DNA behind RNAP, the i-loops are strongly stabilized by the N-tails extending from the surface of the octamer (Figure 7). The future studies will determine the requirement for the N-tails in i-loop formation after transcription further in the nucleosome. 

N-terminal tails of core histones not only affect DNA damage-induced arrest, but also facilitate nucleosomal pausing that is accompanied by i-loop formation (Figure 6 and see Results). Therefore, removal or covalent modifications of histone tails could participate in regulation of nucleosomal pausing in vivo. Indeed, removal of histone tails by the peptidase from the +1 nucleosome participates in the release of Pol II stalled within the +1 nucleosome and in gene regulation [44].

Previously, it has been proposed that i-loops could be used by transcribing Pol II for detection of otherwise undetectable DNA damages in non-template DNA strand [27,30]. Our new data establish that arrest of an RNA polymerase within i-loops can be induced by NT-SSBs localized both in nucleosomal and linker DNA, thus extending implications of this proposed mechanism to internucleosomal linker DNA. Since N-tails of core histones are required for the arrest induced by DNA damages localized within linker DNA, it is also possible that the arrest could be affected or regulated by removal or covalent modifications of the histone tails. The efficiency of the arrest could be modified by cellular factors, such as PARP proteins and DNA repair machinery that are present at the damage site. Furthermore, PARylation of some of these factors by PARP proteins can strongly affect transcription through chromatin [45,46].

## 4. Materials and Methods

### 4.1. Preparation of DNA Templates

#### 4.1.1. Preparation of Templates with Single Nucleotide Gaps at the Positions −10, −11 or −12

Templates with single nucleotide gaps at positions −10, −11 or −12 were obtained using annealing and ligation procedures. 

The promoter part was prepared by the annealing of synthetic purified oligonucleotides (Appendix A). Oligonucleotides (T7A1up1, T7A1dw1 and T7A1dw2; DNA sequences and manufacturers are described in the Table 1) were phosphorylated and radioactively labeled using T4 Polynucleotide Kinase (NEB, Ipswich, MA, USA) following the manufacturer’s instructions. Oligonucleotides were purified using denaturing 8% PAGE with 4M urea and 0.5× TBE. Pairs of oligonucleotides (T7A1up1 and T7A1dw1; T7A1up2 and T7A1dw2) were mixed at equimolar ratios and annealed in the annealing buffer (10 mM Tris HCl, pH 7.5, 100 mM NaCl, 1 mM EDTA), slowly decreasing the temperature from 94 °C to 16 °C. Two obtained solutions were mixed together and annealed by slowly decreasing the temperature from 45 °C to 16 °C. The resulting construct was ethanol precipitated and dissolved in the annealing buffer.

Nucleosomal parts of templates with −10, −11 or −12 gaps were amplified using polymerase chain reaction (PCR). All PCR reactions in this work were conducted using Taq DNA polymerase in 1× Taq DNA polymerase buffer with 3 mM MgCl_2_, 0.2 mM dNTPs and 0.5 µM primers. First, a template DNA fragment was obtained in PCR using Forext603 and 282Revwhst priming oligonucleotides with a template plasmid DNA bearing 603 sequence [34].

The resulting sequence of the first PCR reaction product was:

5′CCAACGCAGCCCAGTTCGCGCGCCCACCTACCGTGTGAAGTCGTCACTCGGGCTTCTAAGTACGCTTAGGCCACGGTAGAGGGCAATCCAAGGCTAACCACCGTGCATCGATGTTGAAAGAGGCCCTCCGTCCTGAATTCTTCAAGTCCCTGGGGTACGGATCCGACG3′ (the 603 nucleosome positioning sequence is underlined).

Then, the obtained PCR product was used as a template in a second PCR reaction with oligonucleotides s603-10 (or s603-11, or s603-12) and 282Revwhst. Products were phenol extracted, ethanol precipitated and digested overnight at 65 °C using TspRI enzyme (NEB). Fragments were purified using the 2% agarose gel with 0.5× TBE and 4M urea, extracted using QIAquick Gel Extraction Kit (Qiagen), phenol extracted and ethanol precipitated. The pellet was dissolved in 10 mM Tris HCl, pH 7.9.

The annealed promoter and TspRI-digested nucleosomal DNA fragment were mixed at equimolar ratios in the annealing buffer and annealed by slowly decreasing the temperature from 45 °C to 16 °C. Products were ethanol precipitated and dissolved in 10 mM Tris HCl, pH 7.9. SSBs were ligated overnight at 16 °C in ligase buffer with T4 DNA ligase (NEB). Ligation was controlled in an agarose gel. Ligation products were purified from the 2% agarose gel, extracted from agarose using QIAquick Gel Extraction Kit, phenol extracted and ethanol precipitated. The pellet was dissolved in 10 mM Tris HCl, pH 7.9. The level of ligation was evaluated on 8% denaturing PAGE (acrylamide:bisacrylamide 19:1, 0.5× TBE and 8M urea).

The resulting sequence of the templates with single strand gaps was:

s603 with gap at position (−10) (s603-10):

5′GATCCCGAAAATTTATCAAAAAGAGTATTGACTTAAAGTCTAACCTATAGGATACTTACAGCCATCGAGAGGGACACGGCGAAAAGCCAACACCGG**CACTG**G^GCAACGCAGCCCAGTTCGCGCGCCCACCTACCGTGTGAAGTCGTCACTCGGGCTTCTAAGTACGCTTAGGCCACGGTAGAGGGCAATCCAAGGCTAACCACCGTGCATCGATGTTGAAAGAGGCCCTCCGTCCTGAATTCTTCAAGTCCCTGGGGTACGGATCCGACG3′

s603 with gap at position (−11) (s603-11) (also schematically shown in Appendix A):

5′GATCCCGAAAATTTATCAAAAAGAGTATTGACTTAAAGTCTAACCTATAGGATACTTACAGCCATCGAGAGGGACACGGCGAAAAGCCAACACCGG**CACTG**G^GCCAACGCAGCCCAGTTCGCGCGCCCACCTACCGTGTGAAGTCGTCACTCGGGCTTCTAAGTACGCTTAGGCCACGGTAGAGGGCAATCCAAGGCTAACCACCGTGCATCGATGTTGAAAGAGGCCCTCCGTCCTGAATTCTTCAAGTCCCTGGGGTACGGATCCGACG

s603 with gap at position (−12) (s603-12):

5′GATCCCGAAAATTTATCAAAAAGAGTATTGACTTAAAGTCTAACCTATAGGATACTTACAGCCATCGAGAGGGACACGGCGAAAAGCCAACACCGG**CACTG**G^GCCCAACGCAGCCCAGTTCGCGCGCCCACCTACCGTGTGAAGTCGTCACTCGGGCTTCTAAGTACGCTTAGGCCACGGTAGAGGGCAATCCAAGGCTAACCACCGTGCATCGATGTTGAAAGAGGCCCTCCGTCCTGAATTCTTCAAGTCCCTGGGGTACGGATCCGACG.

The 603 sequence is underlined and the TspRI site is in bold. The position of single-strand gap is depicted by the ^ symbol.

#### 4.1.2. Preparation of Template with Nick at Position +2

A plasmid containing nucleosome positioning sequence 603 [34] was kindly provided by Dr. Widom. The 603 sequence was modified at five positions to construct the 603+2nick template containing a single site for nicking endonuclease Nt.BsmA1 (NEB) to produce a DNA break in non-template strands after the +2 position (NT-SSB+2) of the nucleosome. To prepare the templates for *E. coli* RNAP transcription, the nucleosome positioning sequence digested by TspRI (NEB) was ligated through the TspRI site to the T7A1 promoter-bearing fragment [37]. The ligated product was cloned in *E. coli* cells and re-amplified with the 5′ end fluorescently labeled primers (ROX-282Forwhst and 282Revwhst) and gel-purified. The following T7A1-s603+2nick template was obtained:

5′CCGGGATCCAGATCCCGAAAATTTATCAAAAAGAGTATTGACTTAAAGTCTAACCTATAGGATACTTACAGCC**ATC**GAGAGGGACACGGCGAAAAGCCAACCCAAGCGACACC*GGCACT*GTCTC^CCGGTTCGCGCGCCCGCCTGCCGAGTGAAATCGTCACTCGGGCTTCTAAGTACGCTTAGCGCACGGTAGAGCGCAATCCAAGGCTAACCACCGTGCATCGATGTTGAAAGAGGCCCTCCGTCCTGAATTCTTCAAGTCCCTGGGGTACGGATCCGACG3′

The 603 sequence is underlined, the TspRI recognition site is in italic, the start site of transcription is in bold and the sugar-phosphate backbone cleavage site for Nt.BsmA1 is indicated with ^. 

NT-SSBs were produced by the incubation of the DNA fragment with Nt.BsmA1 enzyme and the efficiency of the reaction was evaluated by denaturing 8% PAGE. NT-SSBs-containing and intact DNA templates were purified by QIAquick PCR Purification Kit (Qiagen, Hilden, Germany) and assembled into nucleosomes.

#### 4.1.3. Preparation of Template to Obtain EC+2

The 603 sequence was modified at five positions to construct the 603+2stop template allowing stalling of RNAP of *E. coli* at the position +2. To prepare the templates for *E. coli* RNAP transcription, the nucleosome positioning sequence digested by TspRI (NEB) was purified and ligated through the TspRI site to the T7A1 promoter-bearing fragment [37]. The ligated product was cloned in pDS1 vector and re-amplified with radioactively end labeled primers (282Forwhst and 282Revwhst) and gel-purified. The following T7A1-s603+2stop template was obtained:

5′CCGGGATCCAGATCCCGAAAATTTATCAAAAAGAGTATTGACTTAAAGTCTAACCTATAGGATACTTACAGCC**ATC**GAGAGGGACACGGCGAAAAGCCAACCCAAGCGACACCGGCACTGGGGCCAAGTTCGCGCGCCCGCCTGCCGAGTGAAATCGTCACTCGGGCTTCTAAGTACGCTTAGCGCACGGTAGAGCGCAATCCAAGGCTAACCACCGTGCATCGATGTTGAAAGAGGCCCTCCGTCCTGAATTCTTCAAGTCCCTGGGGTACGGATCCGACG3′

The 603 sequence is underlined and the start of transcription is in bold. Details regarding the design of the template and primer sequences will be provided upon request.

### 4.2. Purification of Proteins and Trypsin Cleavage of the Donor Chromatin

Hexahistidine-tagged *E. coli* RNAP was purified as described [47,48]. The GreB protein was purified according to published protocols [49]. Donor chromatin without linker histones was purified from chicken erythrocytes as described [50]. The trypsin cleavage reaction (according to the method [42] with modifications) contained 3.2 mg/mL of chromatin (DNA concentration) and 0.2 mg/mL of trypsin. The reaction was performed in a buffer containing 10 mM Tris-HCl pH 7.5, 0.5 mM EDTA, 350 mM NaCl for 60 min at 25 °C. Aprotinin (0.34 mg/mL) was added to stop the digestion. The extent of digestion was evaluated by Laemmli electrophoresis in 18% SDS–polyacrylamide gel [42].

### 4.3. Nucleosome Assembly

Nucleosomes were assembled on DNA templates by histone octamer transfer from donor chicken erythrocyte chromatin (intact of trypsin cleaved) by dialysis against buffers with decreasing concentrations of NaCl (from 1 M) and analyzed by native PAGE as described [31,51].

### 4.4. Transcription

Transcription by *E. coli* RNAP was conducted as described earlier [27]. Thus, RNAP (200 nM) was incubated with template DNA or nucleosomes (40 nM) in transcription buffer TB40 (20 mM Tris HCl, pH 7.9, 5 mM MgCl_2_, 40 mM KCl and 1 mM beta-mercaptoethanol; the numerical index corresponds to the concentration of KCl, all reagents Sigma-Aldrich, St. Louis, MO, USA) for 7 min at 37 °C to form an open complex. Elongation complexes containing 11-mer RNA (EC-39, the number indicates the position of the active center of the enzyme relative to the promoter-proximal nucleosomal boundary) was formed by addition of 5′-ApUpC and ATP to 20 uM each (all NTPs used GE HealthCare, Chicago, IL, USA) and [α-^32^P]-GTP (3000 Ci/mmol; PerkinElmer, Waltham, MA, USA for −12, −11 and −10 NT-SSBs or Shemyakin and Ovchinnikov Institute of Bioorganic Chemistry, Moscow, Russia for +2 NT-SSB) in TB40 for 10 min at 21 °C (RT). The unlabeled GTP was added to concentration 20 uM and reaction mixture was incubated for 5 min at RT. To prevent multiple rounds of transcription initiation, rifampicin was added to 20 ug/mL. Transcription was resumed by the addition of four NTPs to a final concentration of 200 uM each to pulse-labeled EC-39 in transcription buffer with different concentrations of KCl for the limited time intervals at RT (see text and figure legends for details). Reactions were terminated by phenol extraction (to obtain purified RNA product; Sigma-Aldrich) and ethanol precipitated. RNA markers were obtained using transcription of the intact template in presence of CTP in low concentration (4 uM).

In case of the EC footprinting, CTP was added to 20 uM in TB40 for 10 min at RT to EC-39 to form EC-5 containing 45-mer RNA. Then the EC-5 was washed by ice-cold TB40, TB300 or TB150 using Ni-NTA agarose beads (Qiagen) and CTP, UTP and GTP were added to 20 uM each in TB150 for 10 min at RT to form EC+2. The reaction was terminated by the addition of EDTA (to retain intact ECs for footprinting) or phenol (to obtain purified RNA product). In case of GreB, the enzyme was added at the final step of transcription to different final concentration (see Figures for details).

### 4.5. Hydroxyl Radical Footprinting

Hydroxyl radical treatment of DNA templates, nucleosomes and ECs with a radioactively end-labeled non-template DNA strand was performed as described earlier in the published protocols [52,53]. Conditions were selected to introduce single, randomly positioned NT-SSBs in less than 25% of templates. Level of digestion was controlled by denaturing PAGE. Briefly, footprinting was carried out at a 2.5 μg/mL final concentration of labeled templates in the presence of a 10-fold weight excess of unlabeled −H1 chicken erythrocyte chromatin in TB150 (20 mM Tris HCl pH 8.0, 5 mM MgCl_2_, 2 mM β-ME, 150 mM KCl).

### 4.6. Analysis of RNA Products and DNA Footprinting Probes

Products of transcription and DNA footprinting reactions were analyzed using 8% denaturing PAGE with 8M urea and 0.5× TBE. Sample buffer contained 95% formamide, 0.1% SDS, 0.01% bromophenol blue, 0.01% xylene cyanol, 10 mM EDTA (pH 8.0). Samples were heated for 95 °C for 5 min before loading on the gel. The distribution of products was analyzed using Amersham Typhoon Imager. Calculations of paused RNAP was measured using Fiji ImageJ software [54]. The lanes in the footprinting experiment for Figure 5 were scanned using OptiQuant 5.0 software (Packard Instrument Co., Inc., Meriden, CT, USA) to determine the protected and sensitive regions.

## Figures and Tables

**Figure 1 ijms-24-02295-f001:**
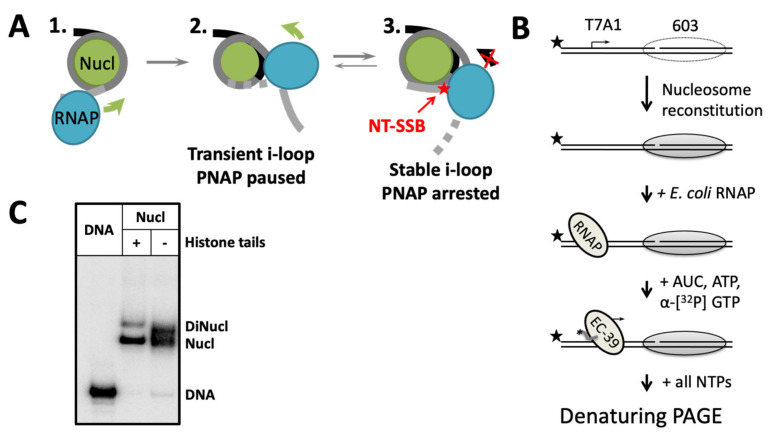
Experimental strategy for analysis of the effect of NT-SSBs on transcription through chromatin. (**A**) A model describing the effect of NT-SSBs on transcription through a nucleosome [27,29,30]**.** As RNAP encounters a nucleosome (intermediate 1), transient intranucleosomal i-loops can be formed (intermediate 2). An NT-SSB positioned behind the transcribing enzyme (shown by red star) greatly stabilizes the i-loop, inducing arrest of RNAP (intermediate 3) [30]. Direction of transcription is indicated by green and black arrows. (**B**) The 603 nucleosomes containing unique ssDNA gap (−12, −11 or −10) or break (+2) in the non-template strand are assembled on the DNA and transcribed for different time intervals in the presence of various concentrations of KCl. Pulse-labeled RNA (the label is shown by black asterisk) is separated by denaturing PAGE. Non-template DNA strand is 5’-labeled (shown by the black stars). Transcription start site is indicated by arrow. (**C**) An example of nucleosomes with ssDNA gap at the position −11 bp in nucleosomal DNA assembled using histone octamers with or without histone tails. DNA was radioactively labeled at 5′-end. Analysis by non-denaturing PAGE.

**Figure 2 ijms-24-02295-f002:**
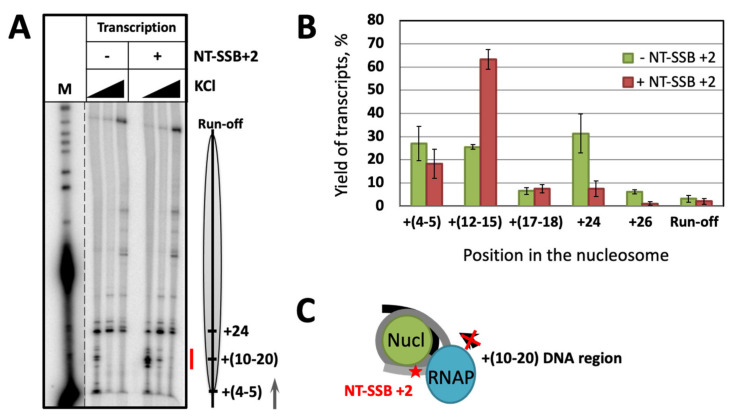
NT-SSB+2 induces arrest of RNAP during transcription of the +(10–20) region. (**A**) Transcription through 603 nucleosomes with and without single ssDNA break at the position +2 by *E. coli* RNAP for 2 min at 40, 150 or 300 mM KCl. Analysis of pulse-labeled RNA by denaturing PAGE. M—pBR322 MspI digest. Position of the nucleosome and direction of transcription are shown by the grey oval and arrow on the right, respectively. The +(10–20) region of RNAP arrest is shown by the red line. (**B**) Quantitative analysis of the RNA products obtained after transcription of the template with and without a nick at the position +2 at 40 mM KCl (averages of 3 experiments and standard deviations are shown). (**C**) The expected structure if the i-loop formed during transcription of the +(10–15) region. NT-SSB+2 is indicated by red star.

**Figure 3 ijms-24-02295-f003:**
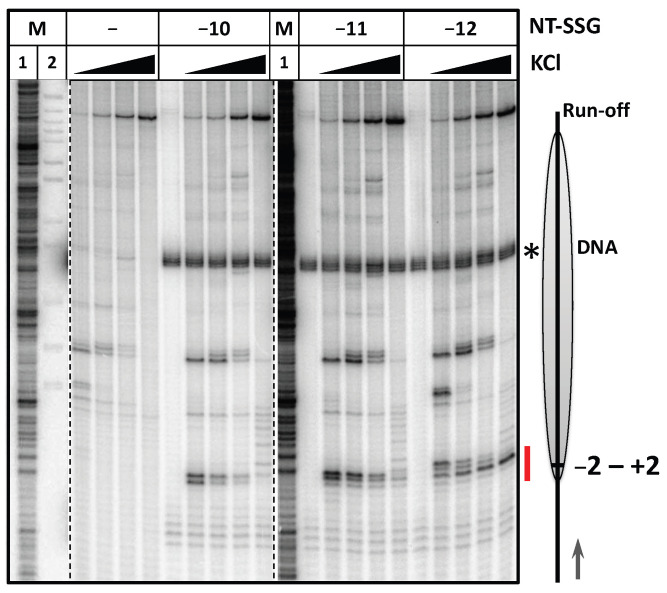
Single strand gaps at positions −10, −11 and −12 induce arrest of RNAP at the nucleosome entry site. Transcription of 603 nucleosomes containing single NT-SSGs at the positions −12, −11 or −10 by *E. coli* RNAP for 5 min at 40, 150, 300 or 1000 mM KCl. Analysis of pulse-labeled RNA by denaturing PAGE. M1—RNA marker (see Methods). M2—pBR322 MspI digest. The radioactively labeled fragment of the gap-containing non-template DNA strand (103 nucleotides in length) is shown by the asterisk. The region of RNAP arrest near position +1 in presence of NT-SSGs is shown by the red line. Other designations as in Figure 2A.

**Figure 4 ijms-24-02295-f004:**
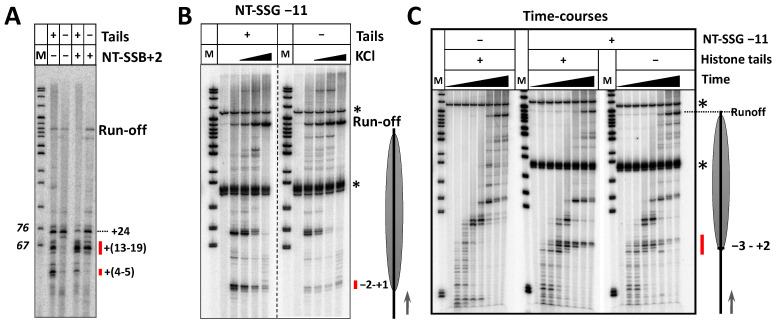
Removal of histone tails results in relief of nick- and gap-induced arrest of RNAP at +(1–5) and +(13–19) regions. (**A**) Nucleosomes NT-SSB+2 with and without histone tails were transcribed by *E. coli* RNAP for 2 min at 40 mM KCl. Analysis of pulse-labeled RNA by denaturing PAGE. M—pBR322 MspI digest. The regions of RNAP pausing/arrest affected by the presence of histone tails are indicated by red lines. (**B**) Nucleosomes with and without histone tails with NT-SSG at the position −11 were transcribed by *E. coli* RNAP for 5 min at 40, 150, 300 or 1000 mM KCl. Analysis of pulse-labeled RNA by denaturing PAGE. M—pBR322 MspI digest. The asterisks indicate radioactively labeled fragments of the DNA template (103 and 273 nucleotides long). Position of the nucleosome and direction of transcription are shown by the grey oval and arrow on the right, respectively. (**C**) Time courses of transcription through the nucleosomes (intact and with NT-SSG-11) with and without tails. The templates were transcribed by *E. coli* RNAP for 1 sec, 2 sec, 4 sec, 8 sec, 1 min, 5 min or 10 min at 150 mM KCl. Analysis of the pulse-labeled RNA by denaturing PAGE. M—pBR322 MspI digest. Radioactively labeled fragments of the DNA template (103 and 273 nucleotides in length) are shown by asterisks. Position of the nucleosome and direction of transcription are shown by the grey oval and arrow, respectively. The region of RNAP pausing near the position +1 in presence of ss-gap is indicated by the red line.

**Figure 5 ijms-24-02295-f005:**
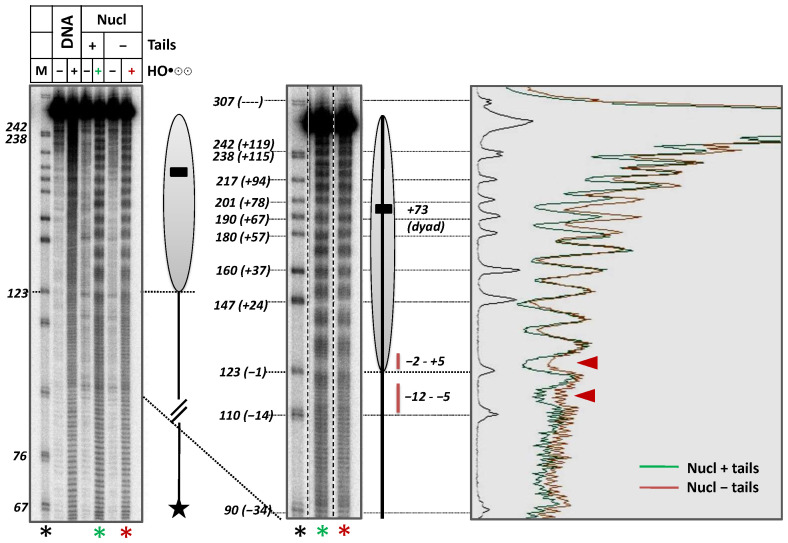
Histone tails strongly contribute to protection of nucleosomal and linker DNA from hydroxyl radicals. The 5′-end of the non-template DNA strand was radioactively labeled. Nucleosomes with or without histone tails were incubated in the presence of hydroxyl radicals. End-labeled DNA was analyzed by denaturing PAGE (on the **left**). M—pBR322 MspI digest. Lanes labeled by colored asterisks (black for markers, green for nucleosomes with histone tails and red for nucleosomes without histone tails) are also shown at a higher magnification in the **middle**; corresponding scans are shown on the **right**. Nucleosome position is shown by the grey oval. Red lines and arrows indicate DNA regions protected from hydroxyl radicals by the histone tails.

**Figure 6 ijms-24-02295-f006:**
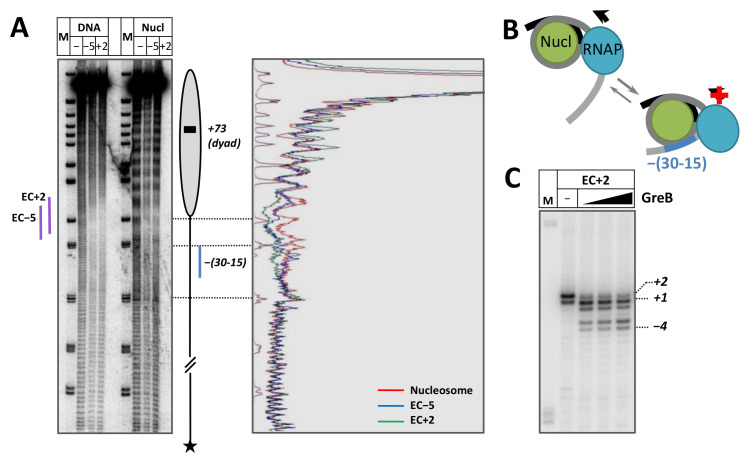
Formation of an i-loop containing backtracked RNAP at the position +1. (**A**) On the left: EC-5 and EC+2 were formed on the 603 DNA and nucleosomes and incubated in the presence of hydroxyl radicals. End-labeled DNA was analyzed by denaturing PAGE (on the left). M—pBR322 MspI digest. Nucleosome position is indicated by grey oval. On the right: Scans of the lanes with nucleosomes and nucleosomal ECs. Purple and blue lines indicate DNA regions protected from hydroxyl radicals by RNA polymerase and DNA region upstream the RNAP protected only in EC+2, respectively. (**B**) EC+2 contains an unstable i-loop. Direction of transcription and arrest of RNAP are indicated by black arrows and red cross, respectively. (**C**) Incubation of EC+2 in the presence of 10, 20 or 40 nM of GreB. Analysis of pulse-labeled RNA by denaturing PAGE. M—pBR322 MspI digest. Positions of active center of RNAP are indicated on the right.

**Figure 7 ijms-24-02295-f007:**
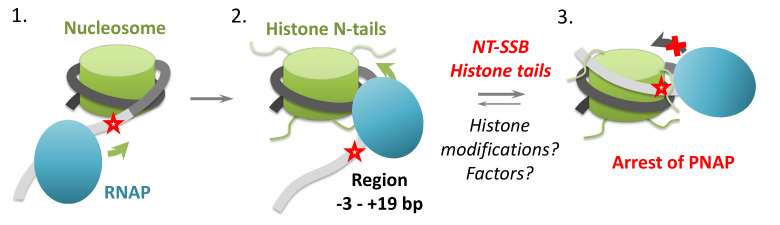
Proposed roles of NT-SSBs and histone N-terminal tails during transcription of the early region of nucleosomal DNA containing a damage. Transcription of the linker and early nucleosomal DNA (intermediate 1) induces transient DNA uncoiling from the histone octamer (intermediate 2) and formation of an unstable intranucleosomal i-loops when the active center of RNA polymerase traverses the −3 to +19 bp DNA region (intermediate 3); the i-loops often incorporate linker DNA. I-loops are stabilized by NT-SSBs (indicated by red stars) and by the histone tails. Histone modifications of the tails and factors interacting with core histones or RNA polymerase could affect the efficiency of loop formation.

**Table 1 ijms-24-02295-t001:** Oligonucleotide sequences and manufacturers.

Name	Oligonucleotide	Manufacturer
Forext603	CCAACGCAGCCCAGTTCGCGCGCCC	IDT *
282Revwhst	CGTCGGATCCGTACCCCAGGGACTT	IDT
s603-10	AAGCCAACACCGGCACTGGGGCAACGCAGCCCAGTTCGCGCGCCCACCTA	IDT
s603-11	AAGCCAACACCGGCACTGGGGCCAACGCAGCCCAGTTCGCGCGCCCACCT	IDT
s603-12	AAGCCAACACCGGCACTGGGGCCCAACGCAGCCCAGTTCGCGCGCCCACC	IDT
T7A1up1	GATCCCGAAAATTTATCAAAAAGAGTATTGACTTAAAGTCTAACCTATAGGATACTTACA	IDT
T7A1up2	GCCATCGAGAGGGACACGGCGAAAAGCCAACACCGGCACTGG	IDT
T7A1dw1	CTATAGGTTAGACTTTAAGTCAATACTCTTTTTGATAAATTTTCGGGATC	IDT
T7A1dw2	GGTGTTGGCTTTTCGCCGTGTCCCTCTCGATGGCTGTAAGTATC	IDT
ROX-282Forwhst	ROX-CCGGGATCCAGATCCCGAAAATTTA	Lumiprobe **

* IDT, Coralville, IA, USA; ** Lumiprobe RUS Ltd., Moscow, Russia.

## Data Availability

The data presented in this study are available on request from the corresponding author.

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
