# Peer review of "Role of Histone Tails and Single Strand DNA Breaks in Nucleosomal Arrest of RNA Polymerase"

_ijms, 2023, doi:10.3390/ijms24032295_

Round 1

Reviewer 1 Report

In this manuscript, Gerasimova et al., studied role of histone tails and SS DNA breaks in nucleosomal arrest of RNA pol.  Detailed background was given in the introduction. Here, Author mentioned about the i-loop is formed in the presence of SSBs position behind the transcribing enzyme, the loop may play a vital role in the transcription coupled repair of DNA . Using this method, can we study  this kind of damage/ repair if its occur during replication coupled or independant chromatin assembly pathway. Results were shown that single NT-SSBs and NT-SSGs localized within the region extend from -12 to +2 bp induced arrest of RNAP during transcription of the region from -3 to +19 bp in the nucleosome.  DNA footprint experiment exhibit the histone tails strongly protect the linker DNA from hydroxyl radicals in intact nucleosomes. Finally, Author have presented evidence  that histone tails and NT-SSBs present in linker DNA strongly make formation of the loops during transcription through proximal region of nucleosomal DNA. 

Over all, Manuscript is well written and every section aptly described with required details.  Therefore,  manuscript should be considered for publication in this journal. 

Author Response

Please see the new cover letter.

Reviewer 2 Report

In this study, the authors aim to build upon their previous study (PESTOV e. al, 2015) and modify their assay to reposition the single-strand break within their substrate to understand RNA Polymerase stalling relative to the position of the break site. While the results are compelling, the manuscript will benefit from the addition of the following points:

·      Describe the physiological relevance of i-loops in a bit more details in the introduction along with a statement on the significance of the study.

·      Add a model/schematic in the end

·      In the cellular setting, how would the PARylation of the SSB affect RNAP pausing?

Author Response

Please see the new cover letter.
